**Data Availability Statement:** Data cannot be shared publicly because of ethical restrictions due

# Survival and predictors of mortality among colorectal cancer patients on follow-up in Hawassa University Comprehensive Specialized Hospital, Sidama region, Southern Ethiopia, 2022. A 5-year retrospective cohort study

**Bargude Balta**[1]*, **Lesley Taylor**[2], **Netsanet Bogale**[1], **Dejene Hailu**[3], **Yasmin A. Zerhouni**[4]

1 Hawassa University Comprehensive Specialized Hospital Cancer Center, Hawassa University College of Medicine and Health Sciences, Hawassa, Ethiopia, 2 Division of Breast Surgery, Department of Surgery, City of Hope Comprehensive Cancer Center, Duarte, California, United States of America, 3 Department of Public Health, Hawassa, Hawassa University College of Medicine and Health Sciences, Hawassa, Ethiopia, 4 Division of Colorectal Surgery, City of Hope, Department of Surgery, Comprehensive Cancer Center, Duarte, California, United States of America

* barjuda@gmail.com

## Abstract

### Background

The incidence and mortality of colorectal cancer were still rising rapidly in many low-income and middle-income countries, which was linked to ongoing societal and economic status. Colorectal cancer is the leading cancer in Ethiopia with relatively lower survival. However, colorectal cancer patients' survival time and predictors have not been well studied in Southern Ethiopia.

### Objective

This study aimed to assess five-year survival and predictors of mortality among colorectal cancer patients at Hawassa Comprehensive Specialized Hospital, Ethiopia.

### Method

Facility-based retrospective cohort study was conducted among 323 patients who visited Hawassa Comprehensive Specialized Hospital from May 1st, 2017 to April 30th, 2022. The Kaplan-Meier survival curve with the Log-rank test was used to estimate the survival time. Bivariable and multivariable Cox proportional hazards regression models were used to determine the net effect of each independent variable on time to death after diagnosis.

### Result

Over the 5-year observation period, the overall mortality rate was 38.5%, with an incidence density of 31 fatalities per 100 person-years observation. Survival at 1, 2, 3, 4, and 5 years

to data contain potentially identifying or sensitive patient information (Even de-identified). Data are available from the authors after appropriate request to the school of public health Research Ethics Committee, pharma college contact: (http://pharma.epua.online/?fbclid=ZzrV4IBLMxJMRvhTd_iHPrcSULUQofRcr8lMsbtOj1w633f6QeRAk73EKXx0N) for researchers who meet the criteria for access to confidential data.

**Funding:** The research project was funded through philanthropic funds to support a graduate student completing a research thesis in the area of colorectal surgery, the results of which are presented as part of the master's thesis. There was no grant number. The funding agencies had no roles in the design of the study; collection, analysis, and interpretation of data and in writing the manuscript.

**Competing interests:** The authors have declared that no competing interests exist.

**Abbreviations:** ACS, American Cancer Society; CEA, Carcinoembryonic Antigen; CHR, Crude Hazard Ratio; DTI, Diagnosis To Treatment Interval; GLOBOCAN, Global Cancer Observatory; IARC, International Agency for Research on Cancer; MRN, Medical Record Number; SEER, Surveillance, Epidemiologic and End Result; SES, Socio-Economic Status; SSA, Sub Saharan African.

was 78%, 53, 32.4%, 23.3%, and 18.7% respectively. The multivariable analysis showed that metastatic disease (AHR = 4.2, CI: 1.5–11.5), baseline carcinoembryonic antigen level ≥5ng/ml (AHR: 2.4, CI: 1.2–5.8), living in rural areas (AHR = 2.2, CI:1.03–4.8) and mucinous carcinoma (AHR = 0.33, CI: 0.13–0.87) were independent predictors of colorectal cancer mortality.

## Conclusion

Overall survival of colorectal cancer patients in the study was low compared to similar studies in developing and developed worlds. A significantly low survival rate was observed for patients with advanced stage, elevated carcinoembryonic antigen levels, and rural residents indicating the key role of early detection and timely initiation of treatment to improve survival and quality of life of patients with colorectal cancer.

## Background

The World Health Organization's Global Cancer Observatory (GLOBOCAN) estimates that there were 19.3 million new cases of cancer and almost 10 million deaths from cancer in 2020. Colorectal cancer (CRC) is one of the commonest types of cancer worldwide, accounting for approximately 10% of all new cancer cases and 8.5% of all cancer deaths [1]. Current diagnostic capacity is poor in Sub-Saharan Africa (SSA), which is a typical cause of lower survival for CRC in SSA countries [2]. The incidence and mortality rates are still rising rapidly in many low-income and middle-income countries, which are linked to ongoing societal and economic status [3, 4].

Overall five-year survival is slightly higher for patients with rectal tumors (67%) than for those with colon tumors (64%), despite generally higher stage-specific survival for colon tumors, because rectal cancer is more often diagnosed at a localized stage (43% vs 38%) [5]. The median survival for colorectal, colon, and rectal cancers in the middle-income countries were 42, 42, and 41 months respectively; while the 1-, 3-, and 5-year relative survival rates ranged from 73.8 to 76%, 52.1 to 53.7% and 40.4 to 45.4% respectively [6]. In Ethiopia, the median survival time of CRC was 21 months, with two-, three- and five-year CRC-specific survival rates of 46.8%, 39.5% and 28.7% respectively. The death of patients diagnosed with CRC In Ethiopia showed a low rate of cancer-specific survival. Histology type, stage of cancer, CEA level at diagnosis, and the type of treatment a patient received significantly determine mortality rate [7].

By comparison, in-country Jordan survival at 5 years was 58.2% [8], and in Saudi Arabian 44.6% [9], and this was driven due to late-stage at diagnosis being the prominent predictor of CRC mortality in Ethiopia [2, 10, 11]. Colorectal cancer is the most common cancer diagnosed in men and the fourth most common in women in Ethiopia [12]. According to a two-year assessment of colorectal cancer patients in Ethiopia, the rectum was the most common site of CRC in 48.3% of cases, followed by the caecum (12.5%), and sigmoid colon (11.5%). More than half of the individuals examined were in stages III to IV of the disease. More than 94% of CRCs in Ethiopia have histologically confirmed adenocarcinoma [13].

Screening programs minimize the incidence of CRC by removing adenomas and can reduce fatalities in detected cancer cases by initiating treatment at an earlier stage [7, 10, 14]. Colorectal cancer is rapidly becoming a major public health issue in Ethiopia with limited therapeutic centers [7]. Its frequency is increasing in Ethiopia, although little is known about

survival rates and predictors of mortality. However, there have been no studies about survival and predictors of mortality of CRC among adult patients in southern Ethiopia. Thus, this study was designed to assess survival and predictors of mortality of CRC among adult patients in Hawassa Sidama Ethiopia, 2022.

## Methods

### Study area

The study was conducted at Hawassa University Comprehensive Specialized Hospital (HUCSH) which is found in Hawassa City, the fifth largest city in Ethiopia. HUCSH is the only hospital serving a population of more than 25 million people in the southern part of Ethiopia. It is the largest and most well-known public hospital in the region. Centers have scarce human resources, scarcity of chemotherapy, and absence of radiation treatment [15].

### Study design

We conducted an institutional-based retrospective cohort study of all colorectal cancer patients treated at HUCSH Oncology Center from May 1, 2017, to April 30, 2022. Data collection was carried out from July 1–30, 2022.

### Population

We included patients who had a complete medical record including a clinical examination that confirmed the presence of a CRC. We excluded patients with an incomplete medical record, carcinoma in situ, unknown clinical stage or histology, other colon conditions that were not CRC, and patients with multiple cancers.

### Sample size determination and sampling

We included all CRC patients during the study period between May 1st, 2017 to April 30th, 2022 in HUCSH.

No sampling procedures were employed since all records were included. In the beginning, 323 colorectal patients, all medical records of a confirmed diagnosis of CRC patients registered from May 1st, 2017 to April 30th, 2022 in HUCSH, were assessed. All study participants who fulfilled the inclusion criteria were identified and included in the study.

### Measurements

We collected baseline characteristics about each patient including age, gender, marital status, place of residence, alcohol consumption, smoking status, age at diagnosis, tumor location, tumor stage, carcinoembryonic antigen (CEA), histology type, tumor grade, chemotherapy, surgery, and delayed treatment defined as starting treatment >60 days after diagnosis. We also characterized their overall health by using the Charleson comorbidity index.

Our primary outcome variable was time to death after diagnosis. We censored patients whose status was unknown, patients who did not develop the primary outcome at the end of the follow-up period, and patients who were lost to follow-up. In this study, survival time was the last date of contact minus the first date of a confirmed diagnosis of colorectal cancer. Patients were contacted during their follow-up visits to the hospital; those who could not be seen in the hospital were contacted via telephone.

## Data collection tools and procedures

The data extraction tool was prepared by using different studies [2, 16–19] and evaluated by oncology experts. Data from the eligible patients' medical records was extracted and imported into the Kobo toolbox. The data were then collected by using ODK collect.

The data archived in the Cancer Registry designed by the Ministry of Health was collected and coded by the International Classification of Diseases (ICD-10), anatomic location of colon cancer (C18), and rectum (C20). To evaluate the primary outcome of death, we looked for the death certificate in the HUCSH cancer registries. In case of absence of the death certificate or other important variables, a telephone interview was done with all patients and/or their caregivers. Then, all charts of colorectal cancer patients, diagnosed between May 1st, 2017 to April 30th, 2022 at HUCSH were retrieved and reviewed Three BSc nurses and one MSc oncology nurse completed the data collection and supervision respectively.

## Data quality assurances

Pre-test on 5% of medical record review was done on a confirmed diagnosis of patients enrolled in 2015 and 2016 two weeks prior to the actual data collection time at Hawassa University Cancer Treatment Centre. As a result, some unrecorded variables were reduced from the data extraction tool. A training guide was prepared. The data collectors and supervisors were trained for two days prior to data collection. Random online evaluation of the recorded data from online ODK extraction application was performed by the principal investigator. Review of data extraction tool filled was gathered and checked for completeness by the principal investigator and supervisors on daily basis. Kobo toolbox was used for data collection to assure the quality.

## Data analysis

Data was anatomized using StataSE 14 (Stata Corp. 2015) for analysis. Basic descriptive analyses were done. The dependent variables were dichotomized into death and censored. A survival table was used to estimate chances of survival after diagnosis of colorectal cancer at different time intervals. Kaplan Meier survival curve, together with the log- rank test was used to estimate the survival time and the presence of a difference in survival among explicatory variables. Before running the Cox Proportional hazard regression model, multi-collinearity diagnosis was checked using variance inflation factors, a value below 10 showing that there is no multicollinearity between two or more predictors. Cox regression was used to find predictors of survival time.

The necessary hypotheticals for the model were checked using goodness of fit test by Schoenfeld residual and variables having P- value>0.05 were considered as fulfilling the supposition. Bivariable Cox regression was fitted and those independent variables that fitted on the bivariable regression lower than or equal to 0.25 position of significance were included in the multivariable analysis. Multiple Cox regression was done at 0.05 level of significance to determine the net effect of each explicatory variable on time to death. The P value lower than 0.05 in the multivariable analysis was considered statically significant. The results of these models were expressed as hazard rates (HRs) with a 95-confidence interval and p- values were used to measure the strength of association and to identify statistically significant predictors.

## Ethics statement

This study approves be the Ethics Committee of Pharma college (Ref: No. PH.09/21). Permission to access patients' records was granted by hospital officials. Verbal consent was obtained

via the telephone from patients or, for patients who had died, from the patient's competent relatives. Patient information was anonymous and kept confidential.

## Results

### Socio-demographic characteristics of the study participants

A total of 323 patients were identified. 63 charts were incomplete and 39 were missing at the time of data collection. A total of 221 study participants were included in the study. The 126 (57%) of the patients were male.152(68.8%) patients lived in rural areas. Around one-third of the study participants were Oromia region. The BMI of more than half of 119 (53.8) of the participants was in the normal range (18.5–24.9 Kg/m$^2$). More than two-thirds of patients were married. Only one-third (67, 30.5%) of patients had public health insurance for medical treatment. More than three-fourths, 121(87.1%) had no history of drinking alcohol. Around 72 (32.6) of the study participants started treatment after 60 days of diagnosis. The mean age at diagnosis for CRC patients was 50.9 years (SD± 13.9). Almost three-fourths of study participants were diagnosed as aged <60 years. The most commonly affected age group is above 50 years (55.3%) and a high incidence of death (60%) occurred in this age group. At the end of the study, 136 patients were living and 85 dead. Further details on patient-related characteristics are presented in Table 1.

### Clinicopathological and treatment related characteristics

More than half 137(62%) of the primary sites of the tumor were found to be colon. Of those patients, 48(35%) died. A large proportion 72.4% of the patients were diagnosed at clinically late stages; locally advanced 50(22.6%) and metastatic 110(49.8). More than one-third 45 (40.9%) of patients diagnosed with metastatic disease died. More than half 126(63.6%) of the tumors were well-differentiated. Most histology 196(87.3%) was adenocarcinoma. Liver and lung were the most common sites of distance metastasis; 37(33.6%) and 32(29.1%) respectively. Concerning the baseline CEA level 90(77.6%) had CEA <5.

The data showed that 125(56.6) patients had ≥2 comorbidities and 59(33) had vascular invention. A total of 150(86.7) patients took only surgical treatment while 104(60.5) were treated with surgery and chemotherapy. CAPOX and FOLFOX are the most common regimes for CRC treatment in the study area (Table 2).

### Overall survival rate of colorectal cancer patients

A total of 221 colorectal cancer patients were followed for 60 months. The overall survival rate was 18.7% at 60 months follow-up (Fig 1). The overall mortality rate for CRC patients over the observation period was 38.5% (95% CI: 32.1–46.3). The estimated cumulative survival rates of colorectal cancer patients at 12, 24, 36, 48, and 60 months were 78%, 53%, 32.4%, 23.3% and 18.7% respectively. The overall median survival time of colorectal cancer patients was 25 months (95% CI: 14–40. The highest mortality rate was found in the first 40 months of confirmed diagnosis of colorectal cancer. The overall death incidence for diagnosed colorectal cancer patients registered at HUCSH during person-year observations was 31 (95% CI:17.7–33.3) deaths per 100 person-years with 3277 months at risk.

### Log-rank survival estimates among predictor variables

The pattern of one survivorship function sitting above another group indicates that the group delimited by the top curve outlives the group restricted by the lower curve. As a result, the presence of any significant variation in survival time was taken into account in this

**Table 1. Socio-demographic characteristics of colorectal cancer patients (n = 221).** SNNPR = South Nations Nationalities and People's Region; DTI = Diagnosis to initiate treatment.

| Variable | Category | Status at last contact | | Total No. (%) |
|---|---|---|---|---|
| | | Death No. (%) | Censored No. (%) | |
| Sex | Male | 50(35.7) | 81(64.3) | 126(57) |
| | Female | 45(42.1) | 40(57.9) | 95(43) |
| **Age of patient** | <40 | 19(33.9) | 37(66.1) | 56(25.3) |
| | 40–49 | 15(37.5) | 25(62.5) | 40(18.1) |
| | 50–59 | 26(46.4) | 30(53.6) | 56(25.3) |
| | 60–69 | 17(34.7) | 32(65.3) | 49(22.3) |
| | ≥70 | 8(40) | 12(60) | 20(9) |
| **Region** | Oromia | 32(40.5) | 47(59.5) | 79(35.7) |
| | Sidama | 29(41.4) | 41(58.6) | 70(31.7) |
| | SNNPR | 24(33.3) | 48(66.7) | 72(32.6) |
| **Residence of patients** | Urban | 28(40.6) | 41(59.4) | 69(31.2) |
| | Rural | 57(37.5) | 95(62.5) | 152(68.8) |
| **Marital status** | Single | 8(50) | 8(50) | 19(8.6) |
| | Married | 70(36.8) | 120(63.2) | 190(86) |
| | Divorced | 4(40) | 6(60) | 10(4.5) |
| | Widowed | 0 | 2(100) | 2(0.9) |
| **Insurance status** | Insurance | 16(23.9) | 51(76.1) | 67(43.8) |
| | Paid | 40(46.5) | 46(53.5) | 86(56.2) |
| **Smoking status** | Smoker | 8(44.4) | 10(55.6) | 18(12.9) |
| | Non- smoker | 41(33.9) | 80(66.1) | 121(87.1) |
| **Alcohol consumption** | Yes | 22(45.8) | 26(54.2) | 48(21.7) |
| | No | 63(36.4) | 110(63.6) | 173(78.3) |
| **DTI** | <60 days | 52(34.9) | 97(65.1) | 149(67.4) |
| | > = 60 days | 33(45.8) | 39(54.2) | 72(32.6) |
| | No | 63(36.4) | 110(63.6) | 173(78.3) |
| **Body mass index** | ≤18.5 | 32(41.6) | 45(58.4) | 77(34.8) |
| | 18.5–24.9 | 41(34.5) | 78(65.5) | 119(53.8) |
| | 25–29.9 | 10(33.3) | 11(52.4) | 21(9.5) |
| | ≥ 30.0 | 2(50) | 2(50) | 4(1.8) |
| **Charles morbidity** | 0 | 10(32.3) | 21(67.7) | 31(14) 65(29.4) |
| | 1 | 33(50.8) | 32(49.2) | 125(56.6) |
| | ≥2 | 42(34.3) | 83(66.4) | |

*0: no comorbidity, 1: only 1 comorbidity, ≥2: 2 and above comorbidity

investigation. The log-rank test data demonstrated that there was a substantial difference in the survival curve for distinct categories (Figs 2–6).

The study found that the median survival time of colorectal cancer with having baseline CEA level greater than 5nm/ml had lower survival than those who had CEA <5nm/ml (26 months CI: 9.2–42.7) as shown by statistical significance with p- value = 0.03. Those colorectal cancer patients who had Adenocarcinoma had better median survival 25(22–27.7) with a static difference of (p<0.001). The median survival time of colorectal cancer patients who were clinically diagnosed at an early stage had better survival at 26 months with a statistical difference of (p<0.001). Survival of patients who were urban residents had better survival compared to those living in rural areas 28 months (22.5–33.5) p-value < 0.05 (Tables 3 and 4).

**Table 2. Clinicopathological and treatment-related characteristics of colorectal cancer (n = 221).**

| Variables | Category | Status at last contact | | Total No. (%) |
|---|---|---|---|---|
| | | Death No. (%) | Censored No (%) | |
| **Primary tumor site** | Colon | 48(35) | 89(65) | 137(62) |
| | Rectum | 37(44) | 47(56) | 84(38) |
| **TMN Stage** (n = 126) | Stage I | 8(38.1) | 13(61.9) | 21(16.7) |
| | Stage II | 8(30.8) | 18(69.2) | 26(20.6) |
| | Stage III | 10(34.5) | 19(65.5) | 29(23) |
| | Stage IV | 27(54) | 23(46) | 50(39.7) |
| **Clinical staging** | Local | 17(27.9) | 44(72.1) | 61(27.6) |
| | Locally advanced | 23(46) | 27(54) | 50(22.6) |
| | Metastatic | 45(40.9) | 65(59.1) | 110(49.8) |
| **Site of distance metastasis** | Liver | 13(35.1) | 24(64.9) | 37(33.6) |
| | Lung | 16(50) | 16(50) | 32(29.1) |
| | Brian | 5(38.5) | 8(61.5) | 13(11.8) |
| | Gastric | 5(45.5) | 6(54.5) | 11(10) |
| | Others | 6 (35.3) | 11(64.7) | 17(15.5) |
| **Tumor grade** (n = 198) | Well differentiated | 45(35.7) | 81(64.3) | 126(63.6) |
| | Moderately differentiated | 17(39.5) | 26(60.5) | 43(21.7) |
| | Poorly differentiated | 13(44.8) | 16(55.2) | 29(14.6) |
| **Histology type** | Adenocarcinoma | 76(30.3) | 120(69.7) | 196(87.3) |
| | Mucinous carcinoma | 9(36) | 16(64) | 25(11.3) |
| **Baseline** | <5 | 29(32.2) | 61(67.8) | 90(77.6) |
| **CEA**(n = 116) | ≥5 | 9(23.7) | 17(65.4) | 26(22.4) |
| **Vascular invasion** (n = 179) | Yes | 21(35.6) | 38(64.4) | 59(33) |
| | No | 49(40.8) | 71(59.2) | 120(67) |
| **Treatment modalities** | Surgery alone | 60(40) | 90(60) | 150(51.7) |
| | Chemotherapy alone | 15(41.7) | 21(58.3) | 36(12.4) |
| | Surgery plus chemotherapy | 39(37.5) | 65(62.5) | 104(35.8) |
| **Mode of surgery**(n = 150) | Elective | 33(34.8) | 53(65.2) | 86(57.5) |
| | Emergency | 24(70.6) | 40(29.4) | 64(42.5) |
| **Type of chemotherapy** | FOLFOX | 23(40.4) | 37(59.6) | 60(42.9) |
| | CAPOX | 21(57.2) | 31(41.8) | 52(37.1) |
| | FOLFIRI | 10(35.7) | 18(64.3) | 28(20) |

Capox: capecitabine, Oxaliplatin; FOLFOX:5-FU, Oxaliplatin. Leucovorin; FOLFIRI:5FU, Leucovorin, Irinotecan. N.B: *Others = (Bone, kidney, small bowel, ovary, gallbladder)

## Predictors of colorectal cancer mortality

In bivariable Cox proportional hazard regression sex, residence, DTI, histologic type, clinical stage, CEA, and Charles comorbidity given were fitted in bivariable analysis at (p < 0.25). Those variables with p-values <0.25 in the bivariable analysis were included in the multivariable analysis. In the multivariable Cox proportional hazards model; clinical stage, histologic type, sex, and baseline CEA level were significant predictors of colorectal cancer mortality (P-value<0.005). In the multivariable analysis, patients with metastatic disease were 4.2 times more likely to die than earlier-stage disease (AHR = 4.2, CI: 1.5–11.5). Patients with Mucinous cell carcinoma was 67% less likely to die earlier (AHR = 0.33, CI: 0.13–0.87) than those with adenocarcinoma. CRC having baseline CEA ng/ml level ≥5 was 2.4 times a higher hazard for

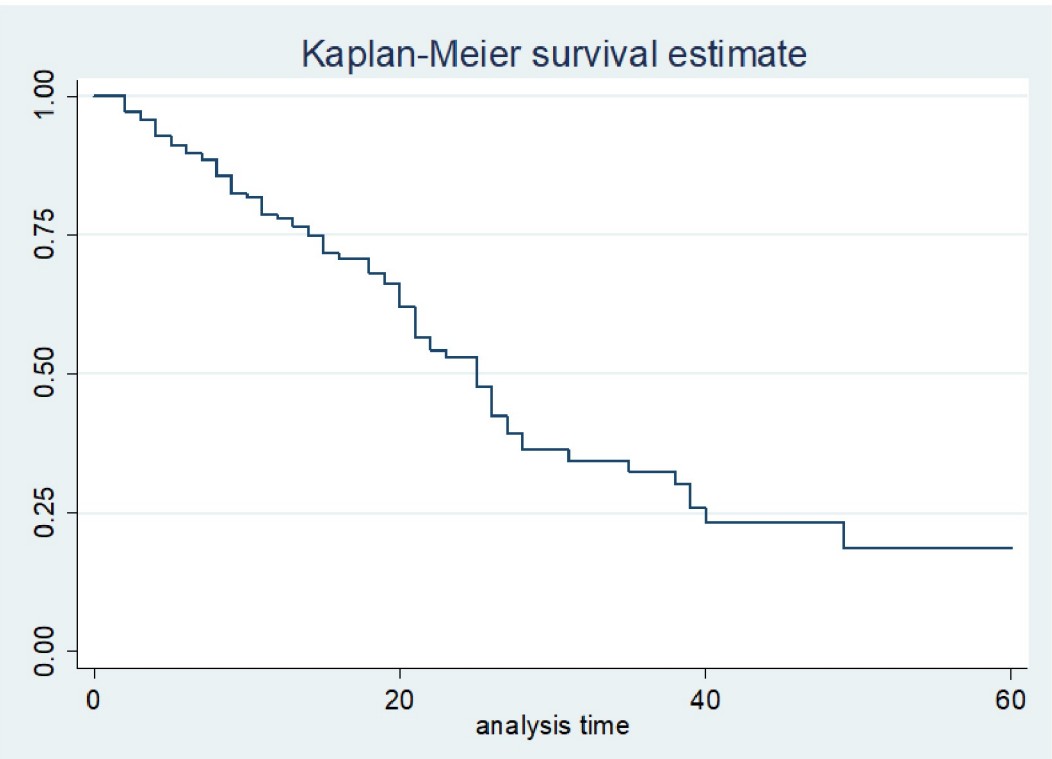

**Fig 1. Overall Kaplan-Meier estimation of survival functions of colorectal cancer patients diagnosed in Hawassa HUCSH, Sidama Region, Ethiopia, from 1st of May 1st, 2017 to April 30th, 2022 (n = 221).**

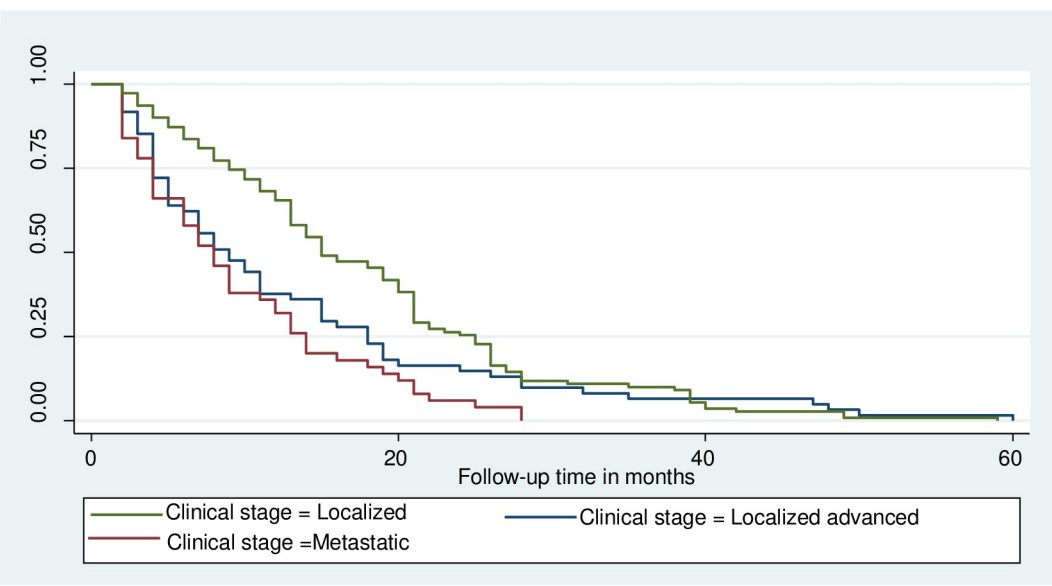

**Fig 2. The Kaplan-Meier failure function compare failure time of colorectal cancer patients with different categories of baseline clinical stage in Hawassa, Sidama, Ethiopia from 1st of May 1st, 2017 to April 30th, 2022 (n = 221).**

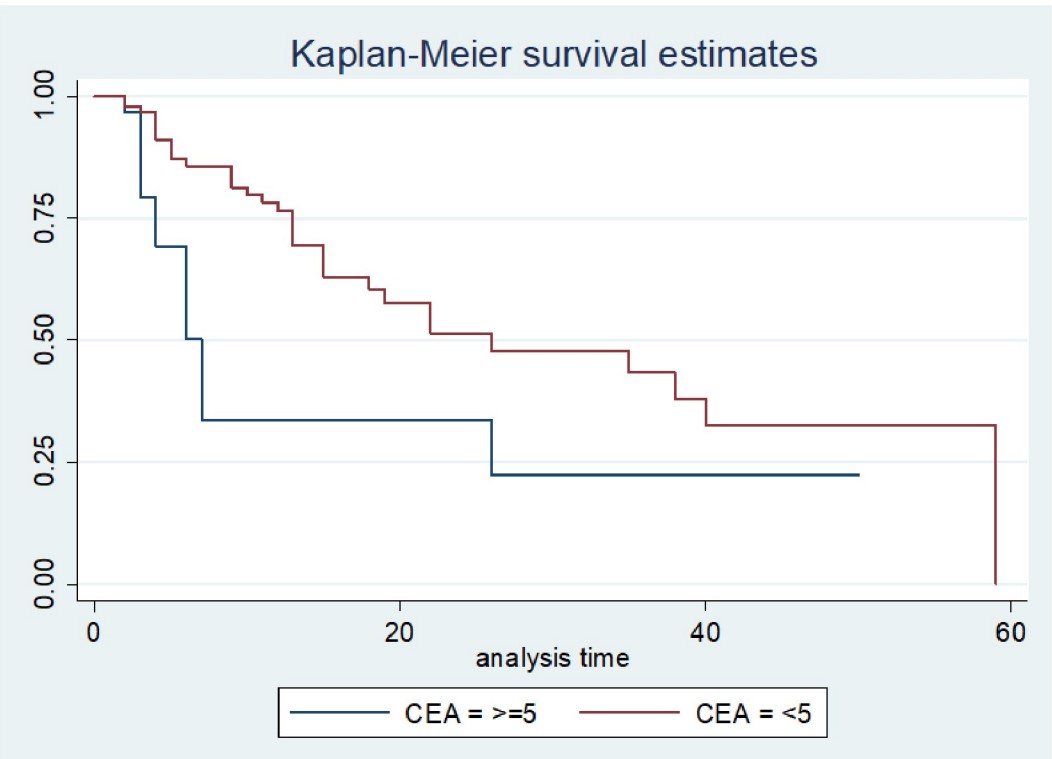

**Fig 3. The Kaplan-Meier failure function compare failure time of colorectal cancer patients with different categories of baseline CEA level in Hawassa, Sidama, Ethiopia from 1st of May 1st, 2017 to April 30th, 2022 (n = 116).**

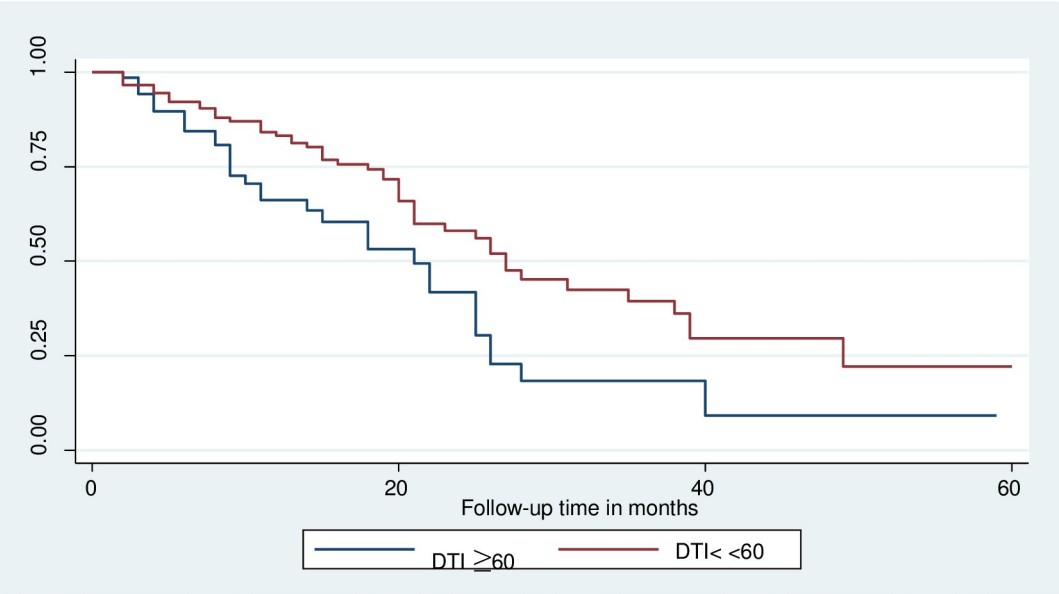

**Fig 4. The Kaplan-Meier failure function compares the failure time of colorectal cancer patients with different categories of DTI in Hawassa, Sidama, Ethiopia from 1st of May 1st, 2017 to April 30th, 2022 (n = 221).** DTI: diagnosis to treatment initiation time.

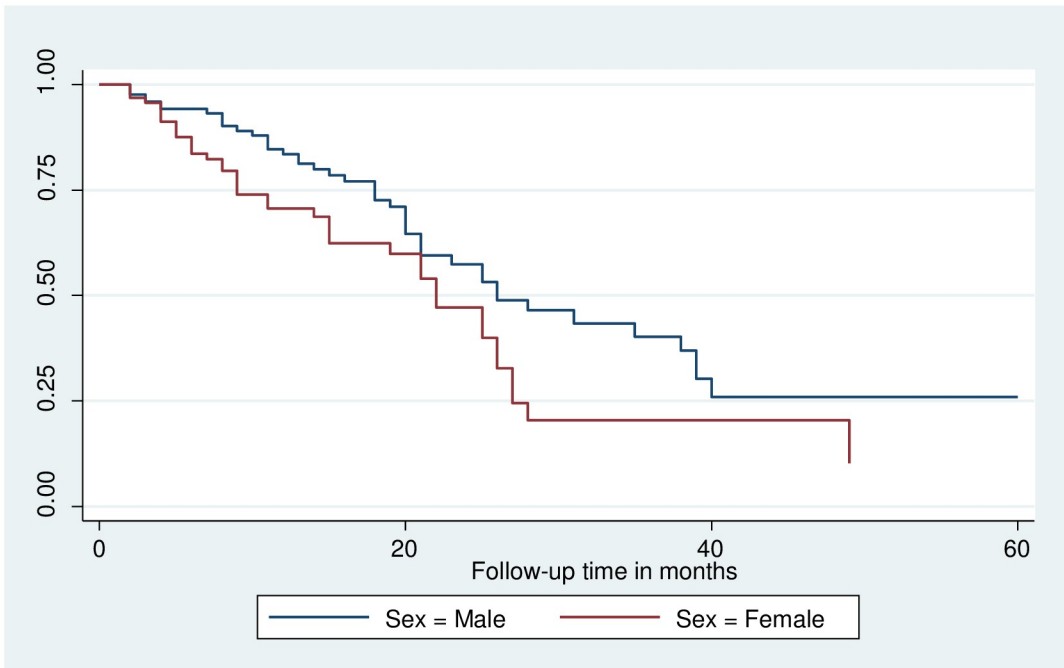

**Fig 5. The Kaplan-Meier failure function compares the failure time of colorectal cancer patients with different categories of sex in Hawassa, Sidama, Ethiopia from 1st of May 1st, 2017 to April 30th, 2022 (n = 221).**

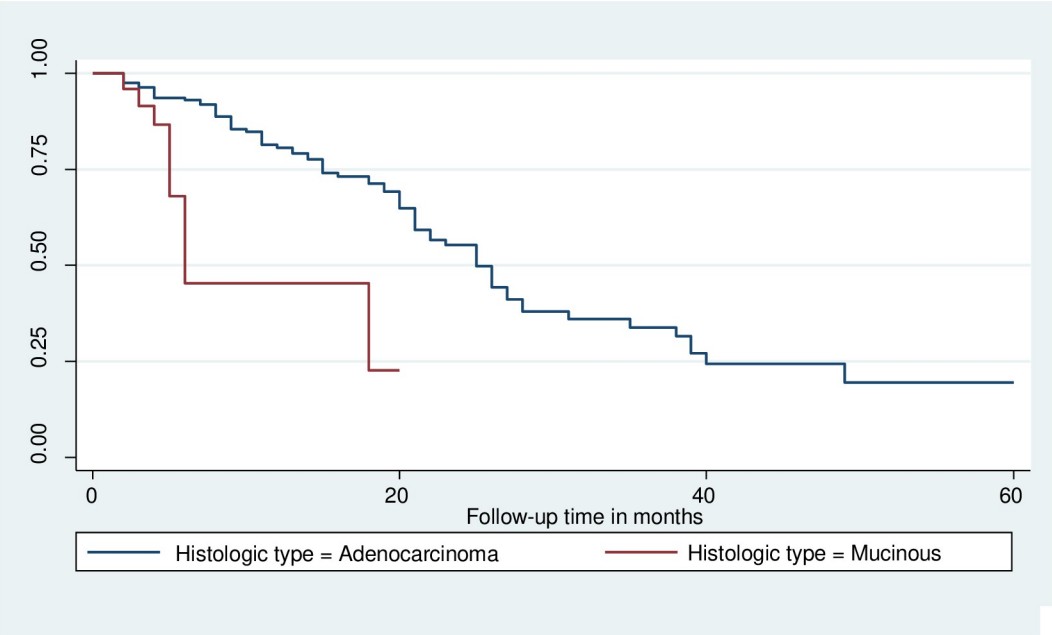

**Fig 6. The Kaplan-Meier failure function compares the failure time of colorectal cancer patients with different categories of histologic characteristics in Hawassa, Sidama, Ethiopia from 1st of May 1st, 2017 to April 30th, 2022 (n = 221).**

**Table 3. Survival time, median survival, cumulative survival probability, and log-rank test for the study population according to patient-related characteristics during the five-year of follow-up (Kaplan- Meier method) of colorectal cancer patients in HUCSH (n = 221).**

| Variable | Category | Median survival in month (95% CI) | 1 year survival (%) | 2-year survival (%) | 3-year survival (%) | 4-year survival (%) | Overall survival (%) | Log-rank test (p-value) |
|---|---|---|---|---|---|---|---|---|
| **Sex** | Male | 26(18.8–33.3) | 83.5 | 57.5 | 40.3 | 30.2 | 25.2 | |
| | Female | 22(17.6–26.4) | 70.6 | 47.2 | 20.4 | 10.2 | - | 0.03 |
| **Age** | <40 | 27(18.5–35.4) | 76 | 52.8 | 47.5 | 42.2 | - | |
| | 40–49 | 26(20.4–30.6) | 79.2 | 74.5 | 39 | 26 | - | |
| | 50–59 | 21(13.2–28.8) | 73.2 | 64.8 | 48.5 | 20.8 | - | |
| | 60–69 | 22(9.7–34.3) | 81.6 | 47.9 | 29.9 | - | 0 | |
| | ≥70 | 25(21.6–28.3) | 90 | 20 | - | 0 | 0 | 0.18 |
| **Residence e** | Urban | 28(22.5–33.5) | 76 | 67.8 | 38.5 | 27 | 18 | |
| | Rural | 21(17.3–24.6) | 78 | 45.5 | 30.7 | 21.9 | - | 0.017 |
| **Marital status** | Single | 27(21.6–32.3) | 71.6 | 30.9 | 15.4 | - | - | |
| | Married | 25(22–27) | 78.6 | 52.7 | 34.7 | 26.1 | 20.9 | |
| | Divorced | 28(8–48) | 67.5 | 45 | 0 | 0 | 0 | 0.5 |
| **Insurance e** | Insurance | 26(21.7–30) | 92.7 | 66.8 | 45.8 | - | - | |
| | Paid | 22(15.9–28) | 73.4 | 46.5 | 25.8 | 6.4 | - | 0.1 |
| **Smoking status** | Yes | 39(18–60) | 72.2 | 62.6 | 31.6 | - | - | 0.6 |
| | No | 23(19.8–26) | 83.8 | 48.1 | 30.2 | 18.1 | - | |
| **Alcohol consumption** | Yes | 25(20–30) | 77.6 | 56.2 | 15.3 | - | - | 0.6 |
| | No | 25(21.7–28) | 78 | 52.7 | 34.2 | 25.4 | - | |
| **BMI** | <18.5 | 23(19.6–26.4) | 76.2 | 48.1 | 19.7 | 13.2 | - | 0.2 |
| | 18.5–24.9 | 26(18.7–33.3) | 80.1 | 58.9 | 38.8 | 34.5 | 25.9 | |
| | 25–29.9 | 21(16.2–25.8) | 73.7 | 49.1 | - | - | - | |
| | ≥ 30.0 | 25(0–32.6) | 33.3 | - | - | - | - | |
| **DTI** | <60 | 27(21.6–32.4) | 83.2 | 58 | 39.4 | 22.2 | - | |
| | ≥60 | 21(14.9–27) | 66.1 | 41.9 | 9.1 | - | - | 0.007 |
| **Charles co morbidity** | 0 | 27(14.5–39) | 83.1 | 50 | 43.1 | - | - | |
| | 1 | 18(10.7–25) | 63.3 | 40.4 | - | - | - | |
| | ≥2 | 27(21–32.5) | 82.8 | 57.4 | 35 | 18.2 | | 0.002 |

DTI: diagnosis to treatment initiated

death than patients with CEA <5 ng/ml (AHR: 2.6, CI: 1.2–5.8). Patients living in rural areas were 2.2(1.03–4.8) times more likely to die of CRC than urban residents (Table 5).

## Discussion

This retrospective cohort study aimed to assess the survival status and predictors of mortality among confirmed diagnosis of colorectal cancer in HUCSH. In the present study the 1-, 2-, 3-, 4, and 5-year survival rates of CRC were 78%, 53, 32.4%, 23.3%, and 18.7% respectively. The results indicated that distance metastasis, elevated baseline CEA level, rural residence, and Histologic characteristics had a significant effect on the survival of CRC patients in the HUCSH cancer center. The present study showed that the majority of CRC patients had early onset of the disease at ages younger than 50 years. This is similar to a previous study done in Tikur Anbessa Hospital Ethiopia indicated a high incidence of early-onset CRC [7]. Similar to the previous findings most study participants were diagnosed at a late stage [17].

**Table 4. Survival time, cumulative survival probability and log-rank test for the study population according to clinicopathological and treatment related characteristics of patients during 5-year of follow-up (Kaplan-Meier method) of colorectal cancer patients in HUCSH (n = 221).**

| Variable | Category | Median survival in month (95% CI) | 1 year survival (%) | 2-year survival (%) | 3-year survival (%) | 4-year survival (%) | Overall survival (%) | Log-rank test (p-value) |
|---|---|---|---|---|---|---|---|---|
| **Primary site** | Colon | 27(24.3–29.8) | 75.2 | 60.6 | 38 | 33.8 | 25.3 | |
| | Rectum | 26(18–23.4) | 82.1 | 42.8 | 25.4 | 16.9 | 8.5 | 0.3 |
| **Baseline CEA** | ≥5 | 7(4.9–9.3) | 53.9 | 39.9 | - | - | - | |
| | <5 | 26(9.2–42.7) | 81.1 | 51.5 | 43.6 | 32.7 | - | 0.03 |
| **Clinical stage** | Early stage | 26 | 74.8 | 54.5 | 36.5 | - | - | |
| | Locally advance | 13(5.4–20.6) | 56.3 | 22.7 | - | - | - | |
| | Metastatic | 16(13.3–28.9) | 47.6 | 39.4 | 22.9 | 13.3 | | 0.00 |
| **Grades of cancer** | Well differentiated | 22(19.3–25) | 77.4 | 44.6 | 27.1 | 21.7 | - | |
| | Moderately | 26(22.2–29.9) | 87.4 | 59.7 | 28 | - | - | |
| | Poorly differentiated | 25(16.8–33.2) | 77.8 | 44.7 | 34.2 | 12.8 | | 0.37 |
| **Vascular invasion** | Yes | 20(12.6–27) | 76.1 | 38.6 | | | | |
| | No | 25(21–29) | 71.4 | 52 | 38 | 28.5 | 21.4 | |
| **Histologic** | Adenocarcinoma | 25(22–27.7) | 80.6 | 55.3 | 33.8 | 24.4 | 19.4 | |
| | Mucinous | 6(0–14.4) | 22.7 | - | - | - | - | 0.000 |
| **Treatment modalities** | Surgery alone | 25(20.1–29) | 89.7 | 76.9 | 57.6 | 57.6 | | |
| | Chemotherapy alone | 22(14–29.9) | 66 | 45.1 | - | - | - | 2.2 |
| | Surgery plus chemotherapy | 25(20.9–29) | 84.1 | 53.4 | 35.1 | 26.3 | 13.2 | |
| **Type of chemotherapy** | FOLFOX | 26(19.3–32.7) | 74.3 | 50.5 | 41.3 | 39 | 17.2 | |
| | CAPOX | 22(16.6–27.5) | 83.6 | 48.9 | 36.7 | 12.2 | - | |
| | FOLFIR | 23(17–28.8) | 83.5 | 45.8 | - | - | 0 | 0.8 |

The current study was lower than the study done in Addis Ababa, Ethiopia 28.7% [7], Jordan 58.2% [8], Saudi Arabiya 44.6% [9], Brazil 63.5% [20], Iran 44% [21], Azerbaijan 46.76% [22] and New Zealand 51% [23]. Survival of colon cancer is 25.3% and rectal cancer is 8.5% in this study; which is lower than previous a study in Brazil presents with rectal tumors (67%) and colon (64%) respectively [5]. However, the overall 5-years survival rate of CRC in this study was slightly higher than Ethiopia 18.1% [17], Ghana at 16% [24] and Cameroon at 12% [25].

In this study, CRC with distance metastasis were 4.2 times more likely to early death which is congruent with studies from different settings, Malaysia [21], and Iran [22]. This is also similar to a previous study done in Addis Ababa, stage four patients were 2.66 times more likely to die of CRC (7). Poorer survival with advanced stages was also seen in a previous study done in Taiwan [6, 26].

There was a significant difference in the survival rate of colorectal cancer based on histologic characteristics, those who were diagnosed with Mucinous cell carcinoma were 67% less likely to die earlier than adenocarcinoma which is congruent with previous findings from a United States study in which Mucinous carcinoma type of CRC had a significant indicator of the outcome as shown the survival rate of 81.4% [27]. This finding is in line with the previous study in Ethiopia Mucinous cell carcinoma were 4.9 times more likely to die from CRC [7].

In this research, rural patients were 2.2 times more likely to die of CRC which is similar to a previous study done in Iran revealed that patients from the rural area had poor survival outcome than town resident patients [21]. It could be due to low living standards or it could be due to more access to treatment and screening in urban areas. According to the log-rank test used in the current study, there is a difference in survival between the sexes, with females having a lower survival rate than males [28].

**Table 5. Results of the bivariable and multivariable cox regression analysis of colorectal cancer patients (n = 221).**

| Variable | Bivariable cHR(95%CI) | P _value | Multivariable aHR (95%CI) | P-value |
|---|---|---|---|---|
| **Clinical stage** | | | | |
| Localized | Ref | | Ref | |
| Locally advance | 0.99(0.6–1.74) | 0.9 | 1.1(0.4–2.5) | 0.8 |
| Metastasized | 2.6(1.5–4.3) | 0.00 | 4.2(1.5–11.5) | 0.006 |
| **Charles comorbidity** | | | | |
| 0 | Ref | | Ref | |
| 1 | 0.85(0.43–1.7) | 0.6 | 0.5(0.1–1.8) | 0.3 |
| ≥2 | 2.1(1.3–3.3) | 0.002 | 2(0.6–7.3) | 0.3 |
| **Histologic type** | | 0.00 | | |
| Adenocarcinoma Mucinous cell carcinoma | Ref | | Ref | 0.024 |
| | 0.24(0.1–0.5) | | 0.33(0.13–0.87) | |
| **Residence** | | | | |
| Urban | Ref | 0.2 | Ref | 0.04 |
| Rural | 1.3(1.1–2) | | 2.2(1.03–4.8) | |
| **Sex** | | | | |
| Female | 0.62(0.4–0.9) | 0.03 | 0.59(0.3–1.2) | 0.8 |
| Male | Ref | | Ref | |
| **Time from Dx to Rx** | | 0.008 | | |
| <60 | Ref | | Ref | 0.2 |
| ≥60 | 1.8(1.2–2.8) | | 0.4(0.1–1.5) | |
| **Baseline CEA** | | 0.04 | | |
| Not elevated(<5g/ml) | Ref | | Ref | 0.02 |
| Elevated(≥5g/ml) | 2.2(1–4.7) | | 2.6(1.2–5.8) | |

Elevated CEA level is an important determinant of CRC in which CEA level ≥5 was 2.3 times a higher hazard to death than patients with those<5 ng/ml (AHR: 2.3, CI: 1.1–5). This finding is congruent with Study findings from Malaysia show that raised CEA levels have a significant effect on the survival of CRC [6]. Also, a similar study done in the USA shows that elevated circulating CEA level reduces 20.7 months of survival in CRC patients [29]. A study done in Tikur Anbessa Hospital, Ethiopia shows similar findings that elevated baseline CEA level >5ng/ml, adjusted Hazard Ratio of 2.3 [7]. According to Jessup et al, the increased tumorigenic potential of CEA-producing tumors could be the reason for the poor prognosis linked with a high baseline CEA level [30].

The lower survival may be attributed to a lack of early screening programs, young cancer treatment facilities, socioeconomic status, and lack of radiation treatments in the current research location. A higher proportion of CRC patients diagnosed in advanced stages lack specialized care and experience a delay in receiving care. Another possible cause is that patients may have delays in diagnosis and treatment due to the high cost and scarcity of pharmaceuticals, as well as their proximity to a healthcare facility are key factors in HUCSH's extraordinarily poor CRC survival rate. Furthermore, the variance could be due to differences in research methodology, particularly the current study's small sample size.

The study has the following strengths: Data were collected by oncologic nurses which had an important role in the quality of the data. It was easy to establish a temporal relationship between outcome death with predictor variables. The current study assessed the effect of treatment options which is not seen in previous studies in Ethiopia.

Despite the above strength this study has the following potential limitations: Incomplete records were excluded so that, the incidence of death may be under or over-estimated. The study was retrospective with a possibility of bias from inaccurate staging of patients, and inaccurate survival and predictors information due to the poor patient chart management. Cause-specific (relative) survival was not determined due to a lack of data on the specific causes of death, this may overestimate the colorectal cancer-related mortality rate. This study does not observe the effect of radiation treatment on survival for CRC due to no documentation of radiation therapy in patients' charts.

## Conclusion

At five-year follow-up, the overall survival probability of colorectal cancer was 18.7%; this finding is lower compared to developed and developing countries. The rate of survival after diagnosis was significantly lower for patients with advanced clinical stage, rural residence, Mucinous cell carcinoma, and elevated CEA level, indicating the critical role of early detection and timely treatment initiation in improving CRC survival and quality of life. We urge that the national cancer control program increase screening services, diagnostic facilities, and timely treatment initiation to improve patients' health and survival rates. Advancing existing cancer centers, and improving the quality of cancer care were important to relieve poor survival. Furthermore, prospective research designs were recommended with a large sample size to reduce the limitations of the current study.

## Acknowledgments

We are grateful to HUCSH-Cancer Center for providing colorectal patient information. The authors would also like to thank the data collectors Mr. Gulema Demissie (MSc), Medu Beshada and Alemayehu Delelegn (MPH)for their outstanding contribution.

## Author Contributions

**Conceptualization:** Bargude Balta, Lesley Taylor, Yasmin A. Zerhouni.

**Data curation:** Bargude Balta, Netsanet Bogale, Yasmin A. Zerhouni.

**Formal analysis:** Bargude Balta, Dejene Hailu.

**Funding acquisition:** Bargude Balta, Lesley Taylor, Netsanet Bogale, Yasmin A. Zerhouni.

**Investigation:** Bargude Balta, Yasmin A. Zerhouni.

**Methodology:** Bargude Balta, Lesley Taylor, Dejene Hailu.

**Project administration:** Bargude Balta, Netsanet Bogale.

**Resources:** Bargude Balta.

**Software:** Bargude Balta, Yasmin A. Zerhouni.

**Supervision:** Bargude Balta, Lesley Taylor, Dejene Hailu.

**Validation:** Bargude Balta, Dejene Hailu, Yasmin A. Zerhouni.

**Visualization:** Bargude Balta, Dejene Hailu.

**Writing – original draft:** Bargude Balta, Dejene Hailu, Yasmin A. Zerhouni.

**Writing – review & editing:** Bargude Balta, Lesley Taylor, Yasmin A. Zerhouni.

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
