## [Decision Letter · Decision Letter 0]

28 Nov 2023

PONE-D-23-23059Survival status and predictors of mortality among colorectal cancer patients at Hawassa University Comprehensive Specialized Hospital, Sidama, Ethiopia, 2022. A five-year retrospective cohort studyPLOS ONE

Dear Dr. Balta,

Thank you for submitting your manuscript to PLOS ONE. After careful consideration, we feel that it has merit but does not fully meet PLOS ONE’s publication criteria as it currently stands. Therefore, we invite you to submit a revised version of the manuscript that addresses the points raised during the review process.

We look forward to receiving your revised manuscript.

Kind regards,

Muhammad Tarek Abdel Ghafar, M.D

Academic Editor

PLOS ONE

2. In the methodology section, Please mention the approval number of the ethics document if applicable.

https://www.deepdyve.com/lp/wiley/colorectal-cancer-statistics-2017-odHoipM3s6

https://www.researchgate.net/publication/340293499_Survival_Status_and_Predictors_of_Mortality_Among_Colorectal_Cancer_Patients_in_Tikur_Anbessa_Specialized_Hospital_Addis_Ababa_Ethiopia_A_Retrospective_Follow-up_Study

In your revision ensure you cite all your sources (including your own works), and quote or rephrase any duplicated text outside the methods section. Further consideration is dependent on these concerns being addressed.

“We are grateful to HUCSH-Cancer Center for providing colorectal patient information. City of Hope Hospital for grant funding. The authors would also like to thank the data collectors Mr. Gulema Demissie (MSc), Medu Beshada and Alemayehu Delelegn (MPH)for their outstanding contribution.”

Reviewers' comments:

Reviewer's Responses to Questions

**Comments to the Author**

1. Is the manuscript technically sound, and do the data support the conclusions?

Reviewer #1: Yes

2. Has the statistical analysis been performed appropriately and rigorously? 

Reviewer #1: Yes

3. Have the authors made all data underlying the findings in their manuscript fully available?

Reviewer #1: Yes

4. Is the manuscript presented in an intelligible fashion and written in standard English?

Reviewer #1: Yes

5. Review Comments to the Author

Reviewer #1: Comments

It is a great opportunity to review this well-articulated paper. I have some comments listed below

1. Methods ‘’Pre-test on 5% of medical record review was done on a confirmed diagnosis of patients enrolled in 2015 and 2016 two weeks prior to the actual data collection time at HUCSH cancer registries’’….Where is the site of pre-test?

2. How to check normality? Multi-collinearity ? please elaborate

3. Results: Charles comorbidity index score ? what does mean 0,1, 2…elaborate….it is score of age, and different comorbidity? Check it again

4. ‘’Patients with adenocarcinoma were 66% less likely to die earlier (AHR=0.33, CI: 0.13-0.87) than Mucinous cell carcinoma’’ ….It is interpreted wrongly……Mucinous adenocarcinoma had 65% less likely to be die as compared with adenocarcinoma…check it from abstract, result, and discussion part

6. PLOS authors have the option to publish the peer review history of their article (what does this mean?). If published, this will include your full peer review and any attached files.

Reviewer #1: No

---

## [Author Response · Author response to Decision Letter 0]

20 Jan 2024

January, 13, 2024 

Muhammad Tarek Abdel Ghafar, M. D

Academic Editor

PLOS ONE

Re: Submission of revised manuscript PONE-D-23-23059

Dear Ms. Muhammad Tarek Abdel Ghafa and Editors, 

We greatly appreciate the reviewers for their constructive comments and suggestions. Please find the attached point-by-point response letter to the reviewers’ comments on the manuscript “Survival status and predictors of mortality among colorectal cancer patients at Hawassa University Comprehensive Specialized Hospital, Sidama, Ethiopia, 2022. A five-year retrospective cohort study” with ID PONE-D-23-23059, for consideration. We have carefully reviewed the comments and we have corrected the manuscript accordingly. We hope that the revised version is now suitable for publication, and we look forward to hearing from you in due course. 

Sincerely, 

Bargude Balta (MSc, MPH) 

Corresponding Author 

Hawassa University college of health science Cancer Center

Hawassa, Ethiopia

Email: barjuda@gmail.com

WhatsApp: +251919739646

Point-by-Point Responses to Reviewers’ Questions and Comments

1. In the methodology section, please mention the approval number of the ethics document if applicable: 

 Response to 1: Thank you for your reminding IRB number. We have adjusted and placed it under ethics statement section of methodology paraphrased as “This study approves be the Ethics Committee of Pharma college (Ref: No. PH.09/21)”. 

Response to 2: We apologize for our mistake; we paraphrased overlapping again

“The funders had no role in study design, data collection and analysis, decision to publish, or preparation of the manuscript.” At this time, please address the following queries:

a) Please clarify the sources of funding (financial or material support) for your study. List the grants or organizations that supported your study, including funding received from your institution. State what role the funders took in the study. If the funders had no role in your study, please state: “The funders had no role in study design, data collection and analysis, decision to publish, or preparation of the manuscript.”

Response to a): The research project was funded through philanthropic funds to support a graduate student completing a research thesis in the area of colorectal surgery, the results of which are presented as part of the master's thesis. There was no grant number. The funding agencies had no roles in the design of the study; collection, analysis, and interpretation of data and in writing the manuscript 

b) If any authors received a salary from any of your funders, please state which authors and which funders.

Response b): The funding for this study was provided by philanthropic donations to the City of Hope Department of Surgery for graduate student research on colorectal cancer in Ethiopia. The lead author of this manuscript, Bargude Balta was the graduate student who used research funding to support research efforts including data collection, supervision of data entry and data processing for Bargude Balta and the research team he assembled in Ethiopia. Salaries were not funded for any of authors.

c) If you did not receive any funding for this study, please state: “The authors received no specific funding for this work.” Please include your amended statements within your cover letter; we will change the online submission form on your behalf.

Response to a: we amended under cover page 

“We are grateful to HUCSH-Cancer Center for providing colorectal patient information. City of Hope Hospital for grant funding. The authors would also like to thank the data collectors Mr. Gulema Demissie (MSc), Medu Beshada and Alemayehu Delelegn (MPH)for their outstanding contribution.”

Please remove any funding-related text from the manuscript and let us know how you would like to update your Funding Statement. Currently, your Funding Statement reads as follows: “The funders had no role in study design, data collection and analysis, decision to publish, or preparation of the manuscript.”

Response to 4: Amendments were done accordingly

6. your Data Availability statement, you have not specified where the minimal data set underlying the results described in your manuscript can be found. PLOS defines a study's minimal data set as the underlying data used to reach the conclusions drawn in the manuscript and any additional data required to replicate the reported study findings in their entirety. All PLOS journals require that the minimal data set be made fully available. 

Response 6: Thank you for your comments we revised Availability of data and materials under declaration section. “Data cannot be shared publicly because of ethical restrictions due to data contain potentially identifying or sensitive patient information (Even de-identified). Data are available from the authors after appropriate request to the school of public health Research Ethics Committee, pharma college contact: (http://pharma.epua.online/?fbclid=ZzrV4IBLMxJMRvhTd_iHPrcSULUQofRcr8lMsbtOj1w633f6QeRAk73EKXx0N) for researchers who meet the criteria for access to confidential data.”.

Point-by-Point Responses to Reviewers’ Questions and Comments

Review Comments to the Author

1. Methods ‘’Pre-test on 5% of medical record review was done on a confirmed diagnosis of patients enrolled in 2015 and 2016 two weeks prior to the actual data collection time at HUCSH cancer registries’’…. Where is the site of pre-test? 

Response to 1: The pre-test was conducted at the same with study site (Hawassa University Cancer Treatment Center located in Hawassa City). This was described on page 4 of the manuscript. Patients charts involvement in pre-test were is a year between May 1st, 2017 to April 30th, 2022 Due to this there is no any duplication of patient card was occurred.

2. How to check normality? Multi-collinearity? please elaborate

Response 2: Thank you for the comment. Normality of quantitative variables was checked using Shapiro-Whilk test, a non-significant result conforming normal distribution. Multi-collinearity diagnosis between the independent variables was check using Variance Inflation Factors where the value below 10 showing that there is no multicollinearity between the independent variables. 

3) Results: Charles comorbidity index score? what does mean 0,1, 2…elaborate….it is score of age, and different comorbidity? Check it again

 Response 3: We apologize again for our unclear description; Charles has also comorbidity score which is based on effect of comorbidity; It has 19 comorbidity conditions was classified based on their weight on survival. Each disease is given a different weight based on the strength of its association with mortality (Charlson et al., 1987): for your references Charles comorbidity classification in the table below.

1 Myocardial infarct, congestive heart failure, peripheral vascular disease, cerebrovascular disease, dementia, chronic pulmonary disease, connective tissue disease, ulcer disease, mild liver disease, diabetes

2 Hemiplegia, moderate or several renal diseases, diabetes with end organ damage, any tumor, leukemia, lymphoma

3 Moderate or severe liver disease

4 Metastatic solid tumor, AIDS

Then we recoded comorbidity index score of “0”, “1” and “≥2” represents: no comorbidity, 1: only 1 comorbidity, ≥2: 2 and above comorbidity

comorbidities, respectively. We coded comorbidity weight 2,3 and 6 to≥2 due to weight of comorbidity in CRC survival and due to small observation in some classification. This was illustrated as a foot note of table 1 on page 6 of the revised manuscript 

4) ‘’Patients with adenocarcinoma were 66% less likely to die earlier (AHR=0.33, CI: 0.13-0.87) than Mucinous cell carcinoma’’ …. It is interpreted wrongly……Mucinous adenocarcinoma had 65% less likely to be die as compared with adenocarcinoma…check it from abstract, result, and discussion part

Response 4: We apologize for our negligence; Thank you again for the comment. This was accepted and corrected in the revised in result and discussion part of manuscript.

---

## [Decision Letter · Decision Letter 1]

20 May 2024

Survival and predictors of mortality among colorectal cancer patients on follow-up in Hawassa University Comprehensive Specialized Hospital, Sidama region, Southern Ethiopia, 2022. A 5-year retrospective cohort study

PONE-D-23-23059R1

Dear Dr. Balta,

We’re pleased to inform you that your manuscript has been judged scientifically suitable for publication and will be formally accepted for publication once it meets all outstanding technical requirements.

Kind regards,

Raffaele Serra, M.D., Ph.D

Academic Editor

PLOS ONE

Additional Editor Comments (optional):

amended manuscript is acceptable

Reviewers' comments:

Reviewer's Responses to Questions

**Comments to the Author**

1. If the authors have adequately addressed your comments raised in a previous round of review and you feel that this manuscript is now acceptable for publication, you may indicate that here to bypass the “Comments to the Author” section, enter your conflict of interest statement in the “Confidential to Editor” section, and submit your "Accept" recommendation.

Reviewer #1: All comments have been addressed

2. Is the manuscript technically sound, and do the data support the conclusions?

Reviewer #1: Yes

3. Has the statistical analysis been performed appropriately and rigorously? 

Reviewer #1: Yes

4. Have the authors made all data underlying the findings in their manuscript fully available?

Reviewer #1: Yes

5. Is the manuscript presented in an intelligible fashion and written in standard English?

Reviewer #1: Yes

6. Review Comments to the Author

Reviewer #1: I would like to express my sincere gratitude to the esteemed authors for their addressing all comments and feedback. The paper has been enriched by incorporating all suggestions, making it a significant contribution to the scientific community regarding the survival status of colorectal cancer in the study area.

7. PLOS authors have the option to publish the peer review history of their article (what does this mean?). If published, this will include your full peer review and any attached files.

Reviewer #1: **Yes: **Atalel Fentahun Awedew
